# Improvement in Solubility and Absorption of Nifedipine Using Solid Solution: Correlations between Surface Free Energy and Drug Dissolution

**DOI:** 10.3390/polym13172963

**Published:** 2021-08-31

**Authors:** Sukannika Tubtimsri, Yotsanan Weerapol

**Affiliations:** Faculty of Pharmaceutical Sciences, Burapha University, Chonburi 20131, Thailand; sukannika@go.buu.ac.th

**Keywords:** nifedipine, solid solution, poorly water-soluble drug, third generation solid dispersion, polysorbate

## Abstract

Ternary solid solutions composed of nifedipine (NDP), amino methacrylate copolymer (AMCP), and polysorbate (PS) 20, 60, or 65 were prepared using a solvent evaporation method. The dissolution profiles of NDP were used to study the effect of the addition of polysorbate based on hydrophilic properties. A solid solution of NDP and AMCP was recently developed; however, the dissolution of NDP was <70%. In the present study, polysorbate was added to improve the dissolution of the drug by altering its hydrophilicity. The suitable formulation contained NDP and AMCP at a ratio of 1:4 and polysorbate at a concentration of 0.1%, 0.3%, or 0.6%. Differential scanning calorimetry and powder X-ray diffraction were used to examine the solid solutions. No peak representing crystalline NDP was observed in any solid solution samples, suggesting that the drug was molecularly dispersed in AMCP. The NDP dissolution from NDP powder and solid solution without PS were 16.82% and 58.19%, respectively. The highest dissolution of NDP of approximately 95.25% was noted at 120 min for the formulation containing 0.6% PS20. Linear correlations were observed between the surface free energy and percentages of dissolved NDP (R^2^ = 0.7115–0.9315). Cellular uptake across Caco-2 was selected to determine the drug permeability. The percentages of cellular uptake from the NDP powder, solid solution without and with PS20 were 0.25%, 3.60%, and 7.27%, respectively.

## 1. Introduction

Nifedipine (NDP) is a typical poorly water-soluble drug that is pharmaceutically used as a calcium channel blocker for the treatment of cardiovascular diseases [1]. Because of its poor water solubility (<5 μg/mL) [2], its dissolution and subsequent oral absorption are limited [3,4]. Therefore, its solubility must be enhanced via various formulations [2,3,4,5]. The term “dispersion” refers to a drug in a polymer matrix or inert carrier [6] as a solid dispersion caused by the use of a meltable compound followed by solvent evaporation or melting solvent methods. A solid dispersion can be a carrier-based process [7], and the preparation of a solid dispersion via a eutectic mixture represents a first generation dispersion [8,9]. In a second generation dispersion, the drug is packaged into amorphous polymeric carriers instead of urea or sugar carriers to overcome the thermodynamic instability of first generation dispersions [10]. Third generation solid dispersions are developed by using surfactants alone or in combination with hydrophilic carriers [11]. Adsorption of the surfactant onto the solid interface can reduce the hydrophobicity of a drug, thereby decreasing the surface free energy at the interface between the liquid and solid. Both excipients and pharmaceutical techniques are important for effective drug delivery [11]. Although the preparation of solid dispersion is reported using modern techniques such as electrospinning [12] electrospraying [13] and fluidized bed technique [14,15], production on a large scale is often difficult for them.

Furthermore, solid solutions are a subset of solid dispersions [11]. A solid solution features a molecularly dispersed drug in hydrophilic carriers, and it can be prepared using solvent evaporation techniques [11]. The dissolution of a drug may be more readily increased by a solid solution than a solid dispersion of small particles in carriers. Solid solutions with polyvinylpyrrolidone K30 and solid dispersions with poloxamer 188 were previously created and compared in an attempt to increase the aqueous solubility of a model hydrophobic drug [15].

All formulations were investigated using differential scanning calorimetry (DSC), powder X-ray diffraction (PXRD), and intrinsic dissolution rates. Solid solutions were found to be an amorphous monophasic system with the transition of crystalline drug to the amorphous state. Solid dispersions were investigated to demonstrate that they retained drug crystallinity and formed a two-phase system. The rates of intrinsic dissolution and concentrations of drug dissolved were higher for that solid solution than for the solid dispersion.

Solid solutions of itraconazole in polyethylene glycol have previously been prepared [16]. The dispersion of itraconazole in melted polyethylene glycol (60–70 °C) did not improve drug dissolution. A solid solution of itraconazole was formed at 120 °C. A solid solution of drug and polyethylene glycol (PEG) at the ratio of 1:2 demonstrated the highest dissolution of itraconazole.

In a study by Minhaz et al. [17], a solid dispersion of clonazepam and amino methacrylate copolymer (AMCP) was prepared via a solvent evaporation method. The results revealed that AMCP decreased the fraction of crystals in the drug. In previous reports, solid dispersions of spironolactone were created using a melting method [18]. A solid dispersion of nimesulide was previously prepared by using polyvinylpyrrolidone K30, PEG 400, and propylene glycol [19]. The dissolution rates of solid dispersions containing surfactants were higher than those of pure drug and solid dispersions without surfactants.

At the water/polymer interfaces, the surface free energy refers to the energy of interaction between the water and polymer [20]. The surface free energy was composed of two kinds of interaction that corresponded to the dispersive component (van der Waals interactions) and the polar component (dipole–dipole interactions and hydrogen bonds) [21]. However, water is a polar liquid, and the polar component of the surface free energy was mainly used to investigate the interaction force at the water/polymer interface. The dispersive component was not reported because it was disproportionate to that of the resulting experiment [20]. Surface free energy is a physicochemical property of a solid that can be assessed by the sessile drop method as a wettability measurement [22]. The relationship between surface free energy and drug release from a matrix containing poly(maleic acid-alt-octadecene) potassium salts (PAM-18K) has been reported [21]. Dissolution decreased with an increase in the hydrophobicity of the polymer (PAM-18K).

The surface properties showed that an increase in PAM-18K in tablets led to an increased contact angle *θ* and a decrease in the surface free energy and polar component, indicating that the hydrophobicity of the tablet surface increased [21]. Research articles published on the investigation of the surface free energy in polymer carriers used for drug delivery are scarce.

In a previous study, solid solutions of NDP and AMCP at the ratios of 1:0.5, 1:1, 1:2, and 1:4 were formulated [23]. However, for a formulation at the ratios of 1:0.5, 1:1, and 1:2, a small endothermic peak and a scattering pattern were observed in DSC and PXRD, respectively. The crystallinity of NDP is dependent on the amount of polymer in the formulation. The solid solution containing AMCP at the ratio of 1:4 showed the absence of the NDP melting peak and scattering pattern, suggesting that NDP was completely soluble in the liquid phase with AMCP. These results indicated that a solid solution was achieved by converting the NDP crystalline form to its amorphous form. Solid solutions of NDP and AMCP at a ratio of 1:4 were suitable; however, the dissolution of NDP was <70%. An insoluble solid carrier, such as silicon dioxide, was added to increase the surface area and improve the dissolution. Complete drug dissolution was achieved at 2 h. In the present study, a solid solution of NDP was developed via a solvent evaporation technique without silicon dioxide according to the formulation mentioned above. Surfactant (polysorbate) was used to improve dissolution of the drug by altering the hydrophilicity of the solid solution. PXRD and thermal analysis were used to characterize the crystallinity of NDP. The Fourier-transform infrared spectroscopy (FTIR) technique was used to investigate the interaction in solid solutions. Surface free energy, and its components, were investigated for all solid solutions. Dissolution profiles (in vitro) were studied to evaluate the performance of the dissolved NDP.

## 2. Material and Methods

NDP (C_17_H_18_N_2_O_6_) was acquired from Xilin Pharmaceutical Raw Material Co. Ltd. (Jintan, Jiangsu, China). AMCP (Eudragit^®^ E PO, C_21_H_37_NO_6_) was obtained from Evonik Industries (Hanau, Germany). Hydrophilic surfactants, namely polysorbate 20 (C_26_H_50_O_10_, HLB 16.7), 60 (C_32_H_62_O_10_, HLB 14.9), and 65 (C_100_H_194_O_28_, HLB 10.5) [24]—hereafter referred to as PS20, PS60, and PS65, respectively, were acquired from P.C. Drug Center Co. Ltd. (Bangkok, Thailand). Caco-2, derived from colorectal adenocarcinoma epithelial cell, was obtained from American type culture collection (catalog no. HTB-37).

### 2.1. Preparation of Solid Solutions

Solid solutions containing the drug and AMCP at a ratio of 1:4 were prepared using a solvent evaporation method as previously described [23]. NDP, polysorbate, and AMCP were dissolved in dichloromethane, and the surfactant (PS20, PS60, or PS65) was added to the solution at various concentrations (0%, 0.1%, 0.3%, and 0.6% referring to each formulation ending with SS, -01, -03, and -06, respectively) (Table 1). Dichloromethane was removed via heating at 40 °C for 72 h. The resulting solid powder was collected and ground using a glass mortar and pestle. After comminution and sifting (60#), the solid solutions were fine yellow particles. The residual dichloromethane analysis was performed to affirm complete elimination of the solvent [25]. Solid solutions lacking dichloromethane were used for further analysis. Table 1 presents the contents of the prepared solid solutions and coded names of the formulations.

### 2.2. Analysis of NDP

High-performance liquid chromatography was used to investigate NDP (HPLC; model Jasco PU-2089 equipped with detector model Jasco UV-2070 plus multi-wavelength UV–vis, Jasco, Tokyo, Japan). According to a prior publication, the separation and analytical method were used [23]. A C18 column was used (5 µm, 4.6 mm × 250 mm) ACE^®^ column. The detector wavelength was 235 nm. To illustrate NDP, an isocratic flow (1 mL/min) condition was used. A mobile phase containing methanol–acetonitrile–water (25:25:50 *v/v*) was filtered via a 0.45 μm nylon membrane filter and degassed in a sonicator bath before use. A 20 μL of sample was injected. NDP peak was computed using ChromNav software (Jasco, Tokyo, Japan). These analyses were performed in triplicate.

### 2.3. Morphological Examination

A scanning electron microscope was used to examine the morphology of the solid solutions (model LEO1450VP, Carl Zeiss Microscopy GmbH, Jena, Germany) with an acceleration voltage of 15 keV. Double-sided adhesive tape was used to adhere the sample powders to a stub. All samples were coated with gold under vacuum before examination.

### 2.4. PXRD

PXRD was used to examine the diffraction peak patterns of all formulations (model MiniFlex II, Rigaku, Tokyo, Japan). Peaks intensity of NDP, solid solution, and physical mixtures (PMs) were investigated. The angular change was 4°/min at an angle of 5–45° 2*θ*. X-ray radiation was delivered using Cu Kα, 30 kV, and 15 mA at a wavelength of 1.5406 Å.

### 2.5. DSC

DSC was used to examine the thermograms of the solid solution, NDP, and PMs (model DSC 8000, Perkin Elmer, Waltham, MA, USA). Each sample (3 mg) was placed in an aluminum DSC pan and sealed. The sample pan was subsequently heated from 30 °C to 200 °C at a rate of 10 °C/min.

### 2.6. Fourier-Transform Infrared Spectroscopy (FTIR)

The interactions between components in solid solutions were investigated using diamond crystal ATR FTIR spectroscopy (Niclolet 6700, Thermo Electron Corporation, Waltham, MA, USA). The samples were placed in an ATR crystal and scanned with a spectral resolution of 2 cm^−1^ in the wave number range 4000–400 cm^−1^.

### 2.7. Hot Stage Microscopy

Hot stage microscopy was used to investigate the properties of solid dispersions. Each sample was placed on a glass slide and covered. The hot stage (model FP82HT, Mettler Toledo, Greifensee, Switzerland) was induced by increasing the temperature at a rate of 1 °C/min. Morphological changes were investigated under an optical microscope (CX41, Olympus, Tokyo, Japan). Polarization of the samples was examined using a polarized filter (CX-AL, Olympus) to investigate the crystals.

### 2.8. Surface Free Energy Determination

The sessile drop method was used to determine the surface free energy (polarity) of solid solutions (n = 3) (model FTA 1000; Data Physics Corporation, San Jose, CA, USA) [26,27]. The glass slides were immersed in solid solutions in dichloromethane, following which they were dried. Residual dichloromethane was removed from the glass slide before the experiment. Polarity was calculated using the surface free energy of the solid solution [28,29]. The contact angles on glass slides were measured using standard liquids with known polar-component and dispersion-component values, such as distilled water (51.0 mN/m and 21.8 mN/m, respectively) and formamide (22.2 mN/m and 36.0 mN/m, respectively) [30] at 25 °C according to the following equations
(1)$$\gamma_{S}=\gamma_{L}\cos\;\theta +\gamma_{LS}$$
(2)$$(1+\cos\;\theta )\;\gamma_{L}=2[2\;\gamma_{S}^{D}\gamma_{L}^{D}/\gamma_{S}^{D}+\gamma_{L}^{D}+2\gamma_{S}^{P}\gamma_{L}^{P}/\gamma_{S}^{P}+\gamma_{L}^{P}]$$
(3)$$\gamma_{S}^{T}=\gamma_{S}^{D}+\gamma_{S}^{P}$$

In the equations,$\;\gamma_{LS}$ is the interfacial tension. $\gamma_{L}\;$ is the surface tension of the liquid. $\gamma_{S}$ is the surface tension of solid. $\gamma_{S}^{T}$ is the total surface free energy of each solid solution on a glass slide, $\gamma_{S}^{P}$ is the polarity of the surface on the solid solution, and $\gamma_{S}^{D}$ is the dispersion force of the surface on the solid solution. The dispersion force and polarity of the standard liquid are represented by $\gamma_{L}^{D}\;$ and $\;\gamma_{L}^{P}$, respectively. The contact angle (*θ*) is the angle made by the liquid and surface of the solid solution.

### 2.9. In Vitro Dissolution Study

Solid solutions were in immediate-release dosage forms. Simulated gastric fluid USP without pepsin (SGF, pH 1.2) at a volume of 900 mL (37 ± 0.5 °C) was chosen as the test medium (USP 2011). A USP dissolution apparatus II (Pharma Test, Berlin, Germany) equipped with a paddle (50 rpm) was used for the study. All samples (10 mg of NDP) were divided for each vessel (n = 3). The sampling times were 5, 10, 20, 30, 60, 90, and 120 min after the initiation of the test. Aliquots (5 mL) of SGF were collected, filtered via a syringe membrane filter (0.45 µm), and analyzed for NDP content using HPLC. A compensatory volume (5 mL) was added to account for the loss of volume after sampling.

### 2.10. Evaluating Cellular Uptake of NDP

Cellular uptake was also studied by using Caco-2 cells. Caco-2 cells were seeded into 24 well plates at density of 2 × 10^4^ cell/well and incubated at 37 °C, 5% CO_2_ for 24 h. The cells were treated with NDP powder, SS, and PS60-06 formulation at the final concentration of 200 µg/mL (below cytotoxicity concentration). After 6 h, treated media were removed and the cells were washed three times with 1 mL phosphate buffer saline (PBS), followed by cell lysis with 1 mL lysis buffer (10 mM Tris-HCL, 150 mM NaCl, 1% TritonX-100, 1 mM EDTA, and 0.1% SDS). After 30 min, 100 µL of cells lysate was gathered into microcentrifuge tube and 900 µL of methanol was then added. The microcentrifuge tubes were centrifuged, and supernatant was collected for further analysis of drug content using HPLC with the same condition in dissolution study.

### 2.11. Stability of NDP

All solid solutions were stored for 3 or 6 months under accelerated (40 °C/75% relative humidity) and ambient conditions (25 °C) to assess the stability of NDP. These experiments were performed in triplicate.

### 2.12. Statistical Analysis

Analysis of variance and Levene’s test for homogeneity of variance were performed by using SPSS version 10.0 for Windows (SPSS Inc., Chicago, IL, USA) [25]. Post hoc testing (*p* < 0.05) for multiple comparisons was performed by using the Scheffé or Games–Howell test if the result of Levene’s test was insignificant or significant, respectively.

## 3. Results and Discussion

### 3.1. Physical Properties of NDP Solid Solution

Solid solutions were prepared as described in Table 1. In the preliminary study, the addition of surfactants at concentrations > 0.6% was attempted; however, soft masses were obtained. Solid solutions containing 0.1%, 0.3%, or 0.6% PS were obtained as fine yellow particles after comminution and sieving (60#). Figure 1 presents the scanning electron microscopy (SEM) images of the solid solutions, polymers, and NDP. The SEM images revealed that NDP had a smooth surface and rectangular shape, whereas AMCP (ground) featured a rough surface and an irregular shape. All solid solutions appeared to have irregular shapes (images for solutions containing 0.1% or 0.3% PS are not shown). NDP seemed to be homogeneously dispersed within the carrier matrix of the solid solution.

A hot stage microscopy was used to examine the microscopic findings of the solid solutions after changing the temperature of NDP and the solid solution. (Figure 2). The images showed that NDP melted at 173 °C, as reported previously [2]. No physical change was observed in AMCP, and similar results were obtained for all solid solutions [23] (some data not shown). These results suggested that a small amount of surfactant (0.1%, 0.3%, or 0.6%) did not change the morphology of solid solutions under temperature alteration.

Figure 3a presents the thermograms of NDP, the solid solutions, and PMs. A high-intensity endothermic peak was observed for NDP powder (173 °C). A low-intensity peak corresponding to NDP was observed in PMs. The results suggest that the peak position of NDP remained unchanged. No peak was found for any solid solution samples (some data not shown). These results corresponded to the images of the hot stage study.

The PXRD peaks are presented in Figure 3b. The characteristic peaks of NDP were observed in the NDP powder samples. The diffraction pattern featured 2*θ* peaks at 8.1°, 10.4°, 11.8°, 19.6°, and 24.6° [31]. In PMs, small peaks of NDP were observed, whereas no peak was observed in any solid solution samples (some data not shown). The results of DSC and PXRD suggested that the drug molecularly disperses in the polymer matrix [32] as a characteristic of solid solution formulations.

In order to clarify the interaction within the system, FTIR was selected to examine the molecular interaction in this research. Figure 4 illustrates the FTIR spectra of AMCP, NDP, SS, PMs, PS20-06, PS60-06, and PS65-06. NDP have remarkable FTIR features with four functional groups at 1225 cm^−1^ due to C–O stretching, 1348 cm^−1^ assigned to NO_2_ symmetric stretching, 1677 cm^−1^ assigned to C=O stretching, and 3326 cm^−1^ due to NH-stretching [1]. AMCP identified a peak of 1238 cm^−1^ for C–N stretching and 1723 cm^−1^ for C=O stretching [33]. For all PMs, the major peak of each component was still observed except for C–N stretching of AMCP, which might be a result of the overlay of strong intensity peak due to C–O stretching of NDP. After incorporating NPD into the solid solutions, the spectra were not significantly different from those found in the PMs. The FTIR spectrums of PS20-06, PS60-06, and PS65-06 were matched to their PMs, which suggested that all components within the system did not form interactions after solid solution fabrication. Furthermore, the same results were found in the formulations with lower surfactant concentrations (0.1% and 0.3%) for all surfactants.

The surface free energy was calculated by determining the contact angle (at 30 s) between droplets of the standard solutions and the solid solution film on the glass slides [31]. As shown in Table 2, solutions containing 0.1%, 0.3%, and 0.6% PS20 exhibited higher surface free energy than those containing PS60 or PS65, as well as the solid solution without surfactants (*p* < 0.05). For the series of solutions containing PS20 and PS60, high surface free energy was achieved by increasing the surfactant content. This effect was not observed in solutions containing PS65. The addition of PS65 reduced the surface free energy. These results suggested that the surface free energy of solid solutions can be improved by the addition of hydrophilic surfactants (PS20 and PS60) but reduced by surfactants with low hydrophilicity (PS65). These results demonstrated that the surface free energy of a solid solution is affected by the type and concentration of the surfactant. From the results regarding high surface free energy, it can be concluded that aqueous adhesion on the solid interface is related to high interaction between aqueous and solid solution interfaces [22].

### 3.2. Dissolution of NDP Solid Solution

An assessment of NDP dissolution was conducted (Figure 5). The dissolution of NDP and the solid solutions after 20 and 120 min is presented in Table 3. At 120 min, the percentages of the drug dissolved from the solid solution without surfactant and NDP powder were 58.19% and 16.82%, respectively. The proportions of NDP dissolved from solutions containing 0.1%, 0.3%, and 0.6% PS20 were 74.62%, 84.26%, and 95.25%, respectively. Meanwhile, the proportions of NDP dissolution from solutions containing 0.1%, 0.3%, and 0.6% PS60 were 64.10%, 70.34%, and 89.95%, respectively (Table 3). On comparing solutions containing the same surfactant concentration, NDP dissolution from formulations containing PS20 was found to be slightly higher than that from those containing PS60. Low dissolution rates were observed for samples containing 0.1%, 0.3%, and 0.6% PS65 (55.01%, 50.87%, and 43.23%, respectively).

These results suggested that a higher NDP dissolution was observed from solid solutions containing AMCP than from unmodified NDP powder [23]. Moreover, the addition of hydrophilic surfactants (PS20 or PS60) enhanced the dissolution of NDP [34], and PS20 was more effective for dissolution than PS60. These results may be attributed to the differences in hydrophilicity between PS20 and PS60 [35]. Conversely, the addition of PS65, which has a lower HLB value than PS20 and PS60, reduced the dissolution of NDP compared with the findings for the solid solution without surfactant. Furthermore, the dissolution of NDP in PS65-containing solutions tended to decrease with increasing PS65 concentrations. Similar results were reported by Minhaz et al. [17], who observed that the addition of a hydrophilic polymer increased drug dissolution. In another study, sucrose laurate (hydrophilic surfactant) was added to gemfibrozil solid dispersions containing PEG 6000 as a carrier [32]. The addition of sucrose laurate could modify the drug dissolution. Furthermore, a meloxicam solid dispersion containing sodium lauryl sulfate reportedly exhibited a significant increase in the rate of dissolution with increasing surfactant concentrations [36]. These findings are consistent with recent results for solid solutions containing PS20 or PS60. The mechanisms responsible for the improved dissolution profiles of solid solutions of NDP and AMCP might include increased wettability and dispersibility. The mechanism that the particles added onto solvents physically separate due to reduced agglomeration has been suggested [37]. The relationship between the surface free energy and drug dissolution at 20 and 120 min is presented in Figure 6. At the same concentration (0.3% or 0.6%) of PS20, PS60, or PS65, linear relationships were observed (R^2^ = 0.7115–0.9315), whereas no such relationships were noted for formulations containing 0.1% of any PS (R^2^ = 0.4183). Similar results were observed with regard to polarity for solutions containing 0.3% or 0.6% of any PS (R^2^ = 0.7160–0.9616, Figure 7) but not for 0.1% PS, possibly because the surfactant concentration was insufficient. The values of dispersion components were obtained from the values of the surface free energy, without the values of polarity components; similar relationships were also observed (data not shown). The findings were consistent with a matrix tablet containing poly(maleic acid-alt-octadecene) potassium salts (PAM-18K), which was reported in the study to reduce the drug release rate when the hydrophobic polymer was added [21]. The wettability of surfactant excipients, as well as their effects on the disintegration and release of plain tablets, was investigated [38]. The results showed that the wetting ability by adding surfactants to the formulation affects the disintegration and release of the tablets. The wetting kinetics of amorphous solid dispersions using various polymers was reported by Verma and Rudraraju [39]. Their research revealed that the improvement of hydrophilicity or liquid spreading of the solid dispersion resulted in superior rates of dissolution. An improvement of dissolution via solid dispersion as indicated by dispersed particles of spironolactone in PEG with varying surfactants (Tween 20, 60, and 80) was previously reported [18]. However, to the best of our knowledge, the relationship of the surface free energy with drug dissolution was newly identified in this study. These current study results suggest that the surface free energy can be routinely investigated for screening the dissolution of solid dispersions to reduce time and material consumption.

### 3.3. Evaluating Cellular Uptake of NDP

Drug permeability has a key role for the achievement of pharmacological effectiveness. Regarding oral administration, the drug amount absorbed across the intestine determines whether the drug concentration in blood circulation is sufficient to have a therapeutic effect on the target organ [40]. In this research, cellular uptake across Caco-2 was selected to determine the drug permeability of the formulation. Figure 8 illustrates the percentage of cellular uptake of the NDP powder, SS and PS20-06 after exposure with each formulation for 6 h. The percentages of cellular uptake were 7.27 ± 0.48, 3.60 ± 0.90, and 0.25 ± 0.19% for PS20-06, SS, and NDP powder, respectively. The highest percentage of cellular uptake was observed in the PS20-06 treatment group with twenty-nine times and two times the amount of the NDP powder and SS, respectively. The result correlated with their in vitro dissolution. The PS20-06 possesses the highest drug dissolution; thus, it generates the highest molecularly dissolved nifedipine, and it is ready to be absorbed across Caco-2 via passive transport [41].

### 3.4. Stability of Solid Solution after Storage

The results of the content analysis of NDP in the selected solid solutions are shown in Table 4. At the end of storage (3 and 6 months), the proportions of NDP content remaining exceeded 99% under both ambient and accelerated conditions. The NDP content remained unchanged after storage. The obtained results for DSC, PXRD, FTIR, and drug dissolution after storage were similar to those prior to storage (data not shown). These results indicate that the developed formulation has good physical stability.

## 4. Conclusions

A solvent evaporation approach was used to fabricate ternary solid solutions containing NDP, AMCP, and polysorbate PS20, 60, or 65. NDP and AMCP were mixed at a ratio of 1:4, with polysorbate at 0.1, 0.3, or 0.6%. The DSC and PXRD results indicated that the drug disperses molecularly in the polymer matrix, which is a characteristic of solid solution formulations. The solubility of NDP was enhanced using a solid solution via the solvent evaporation technique by the addition of a surfactant. The formulation containing 0.6% PS20 had the highest dissolution of NDP, which was approximately 95.25%. The developed formulations (PS20-06) were able to increase cellular uptake more than the surfactant-free (SS) or undeveloped formulations. Surface free energy (polarity) of solid solutions was determined using the sessile drop method. The surface free energy and its components were correlated with drug dissolution. The addition of a surfactant with high hydrophilicity tended to increase the dissolution of NDP, whereas this dissolution was decreased in the presence of a surfactant with low hydrophilicity (PS65).

## Figures and Tables

**Figure 1 polymers-13-02963-f001:**
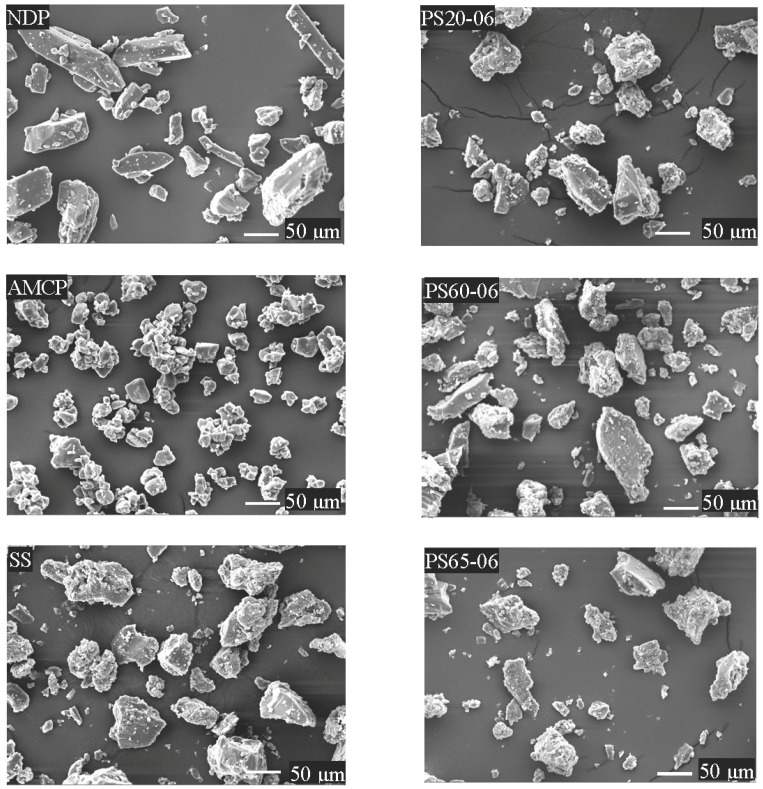
Scanning electron microscopy images (magnification, ×500) of nifedipine, amino methacrylate copolymer, and the solid solutions.

**Figure 2 polymers-13-02963-f002:**
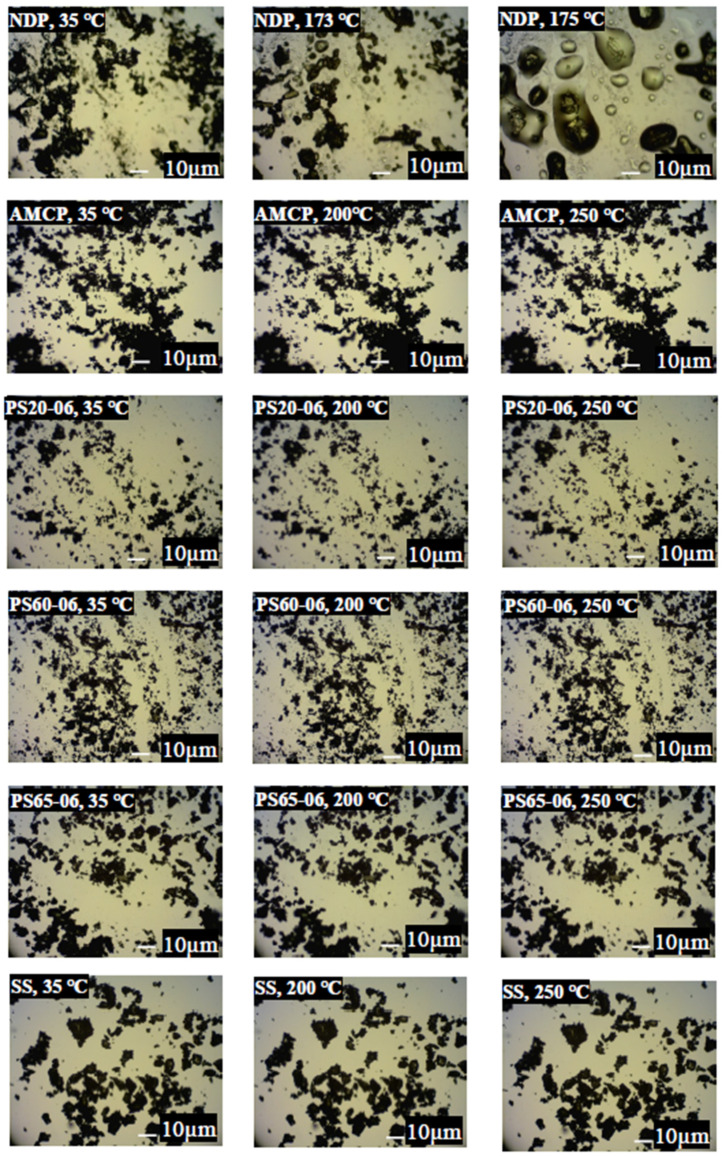
Hot stage microscopy images (magnification, ×400) of nifedipine, amino methacrylate copolymer, and the solid solutions collected at different temperatures.

**Figure 3 polymers-13-02963-f003:**
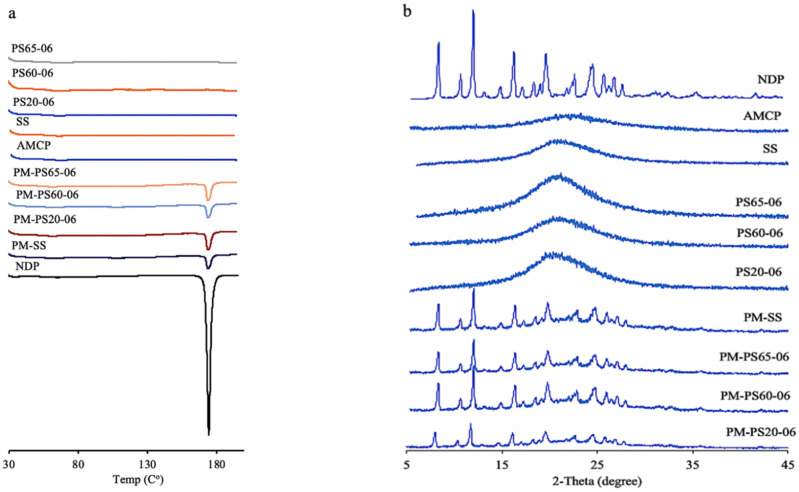
Thermograms (**a**) and X-ray diffraction patterns (**b**) of nifedipine, the solid solutions, and the physical mixtures.

**Figure 4 polymers-13-02963-f004:**
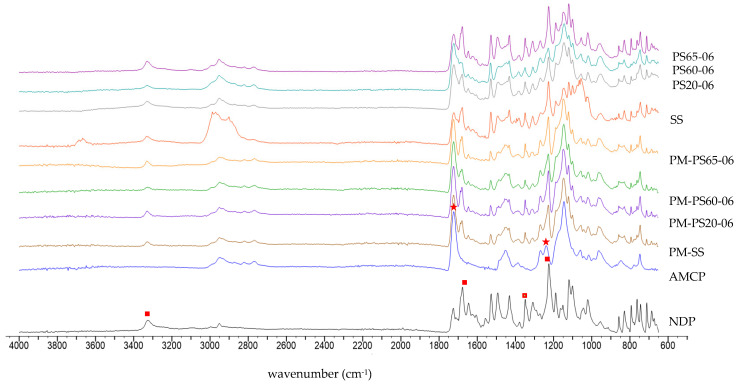
FTIR spectrum of NDP, AMCP, physical mixtures (PMs), SS, PS20-06, PS60-06 and PS65-06: red square represent major functional groups in NDP while red star represent main functional groups in AMCP.

**Figure 5 polymers-13-02963-f005:**
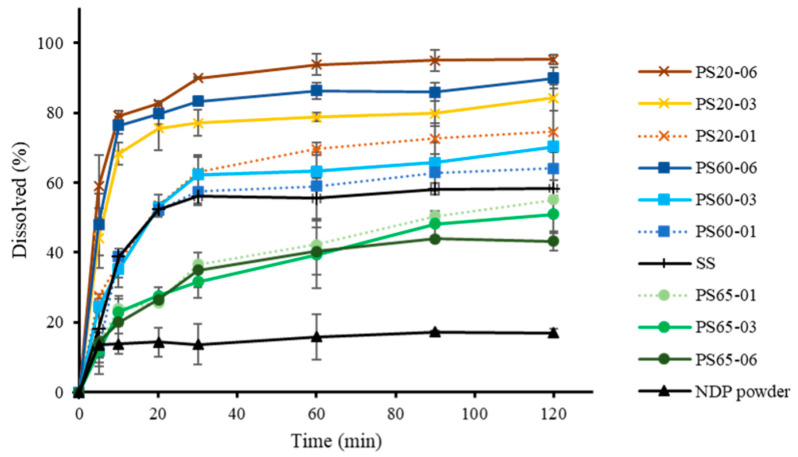
Dissolution profiles of nifedipine powder and solid solutions (error bars indicate standard deviation; n = 3).

**Figure 6 polymers-13-02963-f006:**
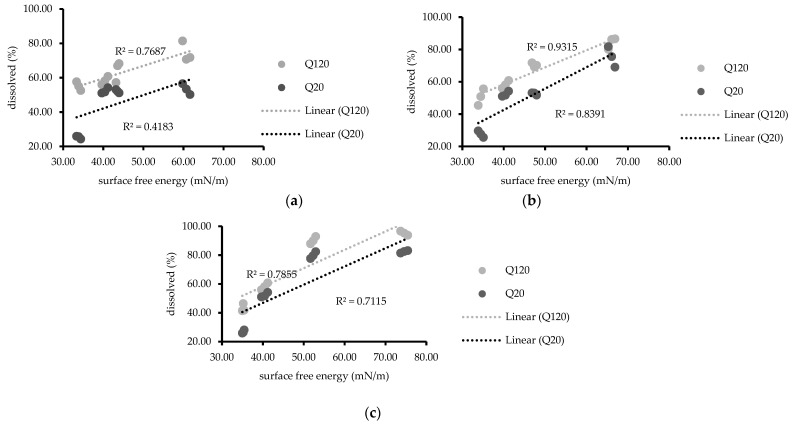
Relationship between surface free energy and percentages of drug dissolved (20 and 120 min) at polysorbate concentrations of (**a**) 0.1%, (**b**) 0.3%, and (**c**) 0.6%.

**Figure 7 polymers-13-02963-f007:**
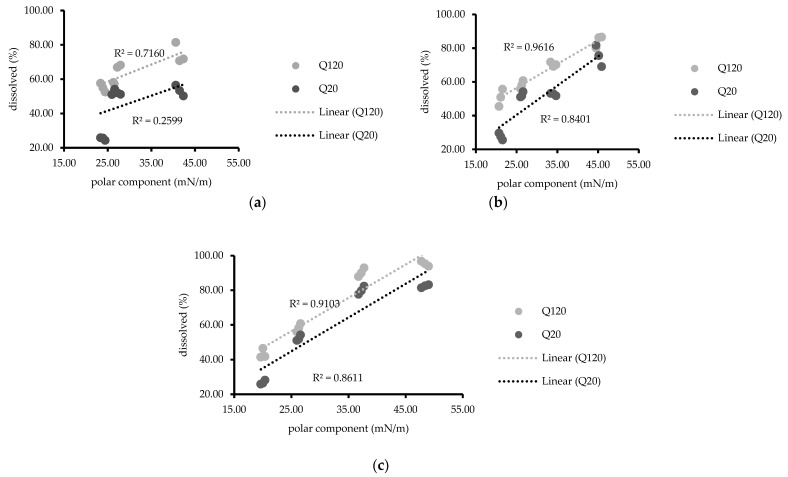
Relationship between the polar component and percentages of drug dissolved (20 and 120 min) at polysorbate concentrations of (**a**) 0.1%, (**b**) 0.3%, and (**c**) 0.6%.

**Figure 8 polymers-13-02963-f008:**
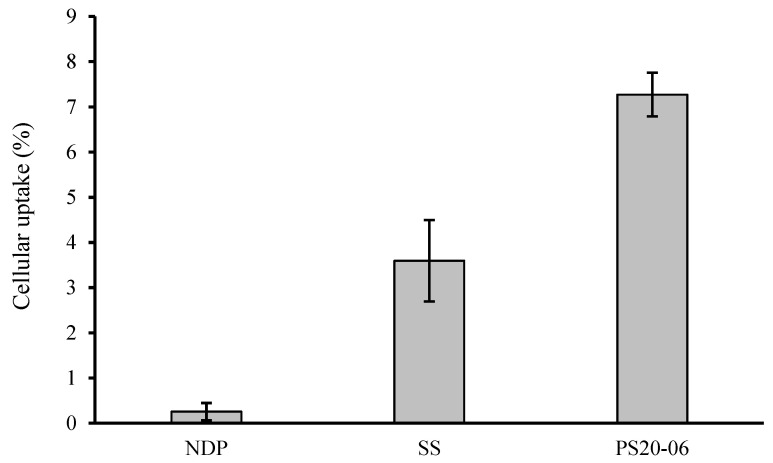
Illustrates the percentage of cellular uptake of NDP powder, SS, and PS20-06 after exposure with each formulation for 6 h (n = 3).

**Table 1 polymers-13-02963-t001:** The content of surfactants, NDP, and polymer in the solid solution formulation.

Formulations	Surfactants (%)	NDP (%)	AMCP (%)
SS	0.00	25.00	75.00
PS20-01	0.10 (polysorbate 20)	24.95	74.95
PS20-03	0.30 (polysorbate 20)	24.85	74.85
PS20-06	0.60 (polysorbate 20)	24.70	74.70
PS60-01	0.10 (polysorbate 60)	24.95	74.95
PS60-03	0.30 (polysorbate 60)	24.85	74.85
PS60-06	0.60 (polysorbate 60)	24.70	74.70
PS65-01	0.10 (polysorbate 65)	24.95	74.95
PS65-03	0.30 (polysorbate 65)	24.85	74.85
PS65-06	0.60 (polysorbate 65)	24.70	74.70

**Table 2 polymers-13-02963-t002:** Surface free energy and its components obtained by sessile drop measurement.

Formulation	Surface Free Energy (mN/m)	Polar Component (mN/m)	Dispersion Component (mN/m)
PS20-01	60.74 ± 0.92 *	41.48 ± 0.87 *	19.26 ± 0.45 *
PS20-03	66.09 ± 0.80 *	45.2 ± 0.68 *	20.89 ± 0.64 *
PS20-06	74.58 ± 0.87 *	48.4 ± 0.63 *	26.18 ± 0.52 *
PS60-01	43.59 ± 0.38 *	27.23 ± 0.72 *	16.36 ± 0.49 *
PS60-03	47.41 ± 0.54 *	34.03 ± 0.66 *	13.38 ± 0.51 *
PS60-06	52.31 ± 0.63 *	37.25 ± 0.48 *	15.06 ± 0.50 *
PS65-01	33.89 ± 0.62 *	23.9 ± 0.54 *	9.99 ± 0.65 *
PS65-03	34.48 ± 0.23 *	21.13 ± 0.43 *	13.35 ± 0.62 *
PS65-06	35.21 ± 0.23 *	20.03 ± 0.37 *	15.18 ± 0.32 *
SS	40.43 ± 0.76	26.3 ± 0.32	14.13 ± 0.30
Glass slide surface	56.31 ± 0.70	31.51 ± 0.69	24.8 ± 0.57

Values are expressed as the mean ± SD (n = 3); * *p* < 0.05 compared with SS.

**Table 3 polymers-13-02963-t003:** Percentages of NDP dissolved after 20 (Q20) and 120 (Q120) min.

Formulations	Q20 (%)	Q120 (%)
NDP	14.41 ± 4.51 ^†^	16.82 ± 1.50 ^†^
SS	51.29 ± 1.52 *	58.19 ± 2.32 *
PS20-01	53.37 ± 3.18 *^,†^	74.73 ± 5.90 *^,†^
PS20-03	75.55 ± 6.33 *^,†^	84.26 ± 3.67 *^,†^
PS20-06	82.52 ± 0.71 *^,†^	95.25 ± 1.47 *^,†^
PS60-01	52.07 ± 0.9 *^,†^	64.10 ± 6.01 *^,†^
PS60-03	52.98 ± 1.03 *^,†^	70.34 ± 1.30 *^,†^
PS60-06	79.67 ± 2.94 *^,†^	89.96 ± 3.04 *^,†^
PS65-01	25.69 ± 1.02 *^,†^	55.02 ± 3.41 *^,†^
PS65-03	27.43 ± 1.79 *^,†^	50.87 ± 5.32 *^,†^
PS65-06	26.43 ± 1.79 *^,†^	43.24 ± 2.76 *^,†^

Values are expressed as the mean ± SD (n = 3), * *p* < 0.05 compared with NDP, ^†^
*p* < 0.05 compared with SS.

**Table 4 polymers-13-02963-t004:** Percentages of NDP dissolved after storage under ambient (25 °C) and accelerated (40 °C/75% relative humidity) conditions.

Formulations	Day 0 (%)	3 Months (%)		6 Months (%)	
		Ambient Condition	Accelerated Condition	Ambient Condition	Accelerated Condition
SS	100.03 ± 0.23	100.02 ± 0.17	100.02 ± 2.37	100.01 ± 0.38	100.00 ± 3.11
PS20-06	100.01 ± 3.27	100.03 ± 4.20	100.02 ± 1.33	100.01 ± 1.27	100.02 ± 0.25
PS60-06	100.03 ± 4.12	100.02 ± 3.25	100.01 ± 1.18	100.02 ± 3.13	100.01 ± 0.13
PS65-06	100.03 ± 2.26	100.02 ± 1.20	100.01 ± 0.33	100.02 ± 1.11	100.01 ± 0.23

Values are expressed as the mean ± SD.

## Data Availability

The data presented in this study are available on request from the corresponding author.

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
