# Peer review of "Improvement in Solubility and Absorption of Nifedipine Using Solid Solution: Correlations between Surface Free Energy and Drug Dissolution"

_polymers, 2021, doi:10.3390/polym13172963_

Round 1
Reviewer 1 Report
The manuscript submitted by Sukannika Tubtimsri and Yotsanan Weerapol reported with the title “Fabrication of a Ternary Solid Solution Composed of Nifedipine, Amino Methacrylate Copolymer, and Polysorbate (Third-generation Solid Dispersion): Correlations between Surface Free Energy and Drug Dissolution”. In this manuscript, there is a major issue which must be fixed before further evaluation and decision of possible publication. I have switched several computers or PDF software, but I cannot see any figures in this pdf manuscript. There are only figure captions attached. Without any valid pictures and figures, it is impossible to read this manuscript and understand the results. Please check the files you uploaded and resubmit the manuscript and supplementary data.
And please check your format as well. The journal "Polymers" has a format requirement and please follow the standard format from this journal's official website.
Author Response
We sincerely apologize for the mistake because we have attached a separate image file. In current revision, the images have been rectified. Could you please reconsider the original manuscript? We have followed the journal's instructions to allow unformatted files to be submitted first. If the original received a review from the reviewer, the journal will request me to submit a formatted file.
Reviewer 2 Report
From a standpoint of polymer science and engineering, the manuscript has no enough merits to support its acceptance for publication. What is more, the submitted manuscript for reviewing has no any Figures to evaluate its results and quality. Thus, I can not recommend its acceptance for publication in its present form.
Provided the authors think that the addition of a third excipient into the polymer-drug composites is a successful point. The authors can submit this job to a journal focusing on pharmaceutics, or re-submit it to POLYMERS withall the Figures in high quality in the manuscript.
Author Response
We sincerely apologize for the mistake because we have attached a separate image file. In current revision, the images have been rectified. Could you please reconsider the original manuscript? We have followed the journal's instructions to allow unformatted files to be submitted first. If the original received a review from the reviewer,the journal will request me to submit a formatted file.
Reviewer 3 Report
This is an interesting paper that requires improvement before it can be considered for publication.
--while the paper has a certificate of editing, this manuscript still requires structural and english grammar improvement
---- avoid the use of "we, our , etc." in the manuscript (line 122, 33, etc)
----avoid the use of consecutive sentences starting with the same word (i.e. "The", etc) throughout. (line 154, line 162-179, etc)
--proper form of references should be used. the current method is confusing. Use proper style, order, etc
--don't lump references. Each reference should be discussed individually
--discuss the novel aspects of this work
--paper structure needs improvement. Use standard titles and have subsections that describe unique aspects of the study.
--need figure to better describe the various experments utilized
--perform a sensitivity amd error analysis for each experiment
--requires a better format for procesure and results
--provide reference for items such as line 231 - spss
figures and ttables need to better integrated into the paper
--line 247 needs an edit ie. )*
--conclusion and results need to be discussed in more detail and conclusions better supported
--12 of 39 references are 5 or fewer years old...perhaps add several more current references
--all figures are missing
--figures and tables should be place in the manuscript
--more detailed titles for tables and figures
Author Response
This is an interesting paper that requires improvement before it can be considered for publication.
Response: Thank you for your consideration and valuable advice.
--while the paper has a certificate of editing, this manuscript still requires structural and english grammar improvement
Response: We appreciate your recommendation. We have submitted a language proof for editing prior to submission of this latest manuscript.
---- avoid the use of "we, our , etc." in the manuscript (line 122, 33, etc)
Response: The sentences have been revised (line 33-34, line 121)
----avoid the use of consecutive sentences starting with the same word (i.e. "The", etc) throughout. (line 154, line 162-179, etc)
Response: The sentences have been revised (line 154-177)
--proper form of references should be used. the current method is confusing. Use proper style, order, etc
Response: The formatted for reference has been corrected in current revision.
--don't lump references. Each reference should be discussed individually
Response; Thank you for recommending us. The reference in the sentence has been modified in this manuscript.
--discuss the novel aspects of this work
Response: An important discussion point is found the new relationship of the surface free energy vs drug dissolution. We attempted to find the cited documents and discussed all relevant parts in this manuscript. (line 319-333) We expect this new relationship to be a method for faster evaluation of drug dissolution as we have written in the manuscript. (line 338-342)
--paper structure needs improvement. Use standard titles and have subsections that describe unique aspects of the study.
Response: We appreciate the reviewer's recommendation and we have edited the titles as follows.
“Improvement in Solubility and Absorption of Nifedipine using Solid Solution: Correlations between Surface Free Energy and Drug Dissolution”
--need figure to better describe the various experments utilized
Response: We apologize for the mistake. Because we have attached a separate image file. In current revision, the images have been rectified.
--perform a sensitivity amd error analysis for each experiment
Response: We apologize for the previous mistake. Error and sensitivity information are added in the figures.
--requires a better format for procesure and results
Response: We appreciate your valuable advice; we have added information to the new graphical abstract in this revision.
--provide reference for items such as line 231 – spss
Response: the reference has been added in this revision. (line 239)
figures and tables need to better integrated into the paper
Response: We sincerely apologize for the mistake. Because we have attached a separate image file. In current revision, the images have been rectified.
--line 247 needs an edit ie. )*
Response: The sentences have been revised (line 255-256)
--conclusion and results need to be discussed in more detail and conclusions better supported
Response: The discussion and conclusion have been modified. (line 327-333 and line 365-378)
--12 of 39 references are 5 or fewer years old...perhaps add several more current references
Response: Thank you for recommending us. The references in the sentences have been modified in this manuscript.
--all figures are missing
Response: In current revision, the images have been rectified.
--figures and tables should be place in the manuscript
Response: We sincerely apologize for the mistake. In current revision, tables and figures have been placed in main file.
--more detailed titles for tables and figures
Response: We sincerely apologize for the information contained in the missing document. This manuscript has been added and modified.
Round 2
Reviewer 1 Report
The manuscript submitted by Sukannika Tubtimsri et al reported on the “Fabrication of a Ternary Solid Solution Composed of Nifedipine, Amino Methacrylate Copolymer, and Polysorbate (Third-generation Solid Dispersion): Correlations between Surface Free Energy and Drug Dissolution”
In this work, NDP, AMCP, and polysorbate within different ratios were prepared and studied. I believe the manuscript was well arranged and organized and it depicts an important modification in NDP drug delivery.
There are some minor changes I suggest the authors to make.
In Figure 7, because the columns were totally black, I cannot see the bottom part of deviation bars clearly. Please fix it.
In many tables, PSxx-01/ PSxx-03/ PSxx-06 are showed multiple times. Please explain those number in the main text. I don’t quite understand why they were labeled as -01/-03/-06.
In figure 5 and 6, the X axis only has units labeled. Please show the name of X axis.
Author Response
The manuscript submitted by Sukannika Tubtimsri et al reported on the “Fabrication of a Ternary Solid Solution Composed of Nifedipine, Amino Methacrylate Copolymer, and Polysorbate (Third-generation Solid Dispersion): Correlations between Surface Free Energy and Drug Dissolution”
In this work, NDP, AMCP, and polysorbate within different ratios were prepared and studied. I believe the manuscript was well arranged and organized and it depicts an important modification in NDP drug delivery.
Response: Thank you for your consideration and valuable advice.
There are some minor changes I suggest the authors to make.
In Figure 7, because the columns were totally black, I cannot see the bottom part of deviation bars clearly. Please fix it.
Response: The Figure 7 has been modified.
In many tables, PSxx-01/ PSxx-03/ PSxx-06 are showed multiple times. Please explain those number in the main text. I don’t quite understand why they were labeled as -01/-03/-06.
Response: We appreciate your recommendation; more detail of formulation codes have been added. (line 152-155)
In figure 5 and 6, the X axis only has units labeled. Please show the name of X axis.
Response: The chart has been rectified. (Figure 5,6)
Reviewer 2 Report
The dissolution of poor water-soluble drugs comprise one of the most difficult and important issues in the fields of drug delivery and medicated materials. The present study is a successful case study on promoting the dissolution of nifedipine (NDP), in which amino methacrylate copolymer (AMCP), and polysorbate (PS) were utilized as the carriers. Thus, these contents are interesting and fall well within the scope of POLYMERS. However, it’s quality can be improved. I recommend its acceptance after the following issues are well addressed.
- The background is narrow in knowledge! Any kinds of solid dispersions have two supports, i.e. raw drug carriers and preparation methods. The authors please add some most recent developments for the readers, i.e. both excipients and pharmaceutical techniques are important for effective drug delivery ( Drug Deliv. 2021, 18, 2–3). Meanwhile, solvent evaporation methods should be simply concluded in several sentences such as new methods are always introduced in the preparation of solid dispersion, such as electrospinning (https://doi.org/10.1007/s40242-021-1006-9), electrospraying (Materials & Design, 2018, 143, 248-255) and fluidized bed technique (https://doi.org/10.3390/polym12112623; https://doi.org/10.3390/polym12020400).
- The molecular formula of the three raw components please be enclosed in the manuscript to explain the potential secondary interactions among the components for forming the solid solution, or molecular solid dispersion or amorphous solid dispersions.
- It should be better to conduct the FTIR experiments to disclose the compatibility among these components.
- The writing can be improved, e.g. line 324 pow der, and so on.
- The references are too old, only 5 of them are within the most recent three years. To relate your job with the most recent developments can benefit a high impact of your article after publication. And the formats are chaotic, upper and lower cases in the references’titles and authors’ names being full or “et al”.
Author Response
The dissolution of poor water-soluble drugs comprise one of the most difficult and important issues in the fields of drug delivery and medicated materials. The present study is a successful case study on promoting the dissolution of nifedipine (NDP), in which amino methacrylate copolymer (AMCP), and polysorbate (PS) were utilized as the carriers. Thus, these contents are interesting and fall well within the scope of POLYMERS. However, it’s quality can be improved. I recommend its acceptance after the following issues are well addressed.
Response: Thank you for your consideration and valuable advice.
The background is narrow in knowledge! Any kinds of solid dispersions have two supports, i.e. raw drug carriers and preparation methods. The authors please add some most recent developments for the readers, i.e. both excipients and pharmaceutical techniques are important for effective drug delivery ( Drug Deliv. 2021, 18, 2–3). Meanwhile, solvent evaporation methods should be simply concluded in several sentences such as new methods are always introduced in the preparation of solid dispersion, such as electrospinning (https://doi.org/10.1007/s40242-021-1006-9), electrospraying (Materials & Design, 2018, 143, 248-255) and fluidized bed technique (https://doi.org/10.3390/polym12112623; https://doi.org/10.3390/polym12020400).
Response: We appreciate your recommendation; the introduction has been revised and new references have been added.We tried to find one of the references you suggested but couldn't find it. (Drug Deliv. 2021, 18, 2–3) (line 72-75)
The molecular formula of the three raw components please be enclosed in the manuscript to explain the potential secondary interactions among the components for forming the solid solution, or molecular solid dispersion or amorphous solid dispersions.
Response: We appreciate your recommendation; more detail have been added. (line 142-145)
It should be better to conduct the FTIR experiments to disclose the compatibility among these components.
Response: Thank you for giving us advice. FTIR experiments have been performed (line 189-193), the results of the experiment and discussion were revised in this manuscript. (line 281-294)
The writing can be improved, e.g. line 324 pow der, and so on.
Response: We appreciate your recommendation; the sentences have been modified. (line 348-354)
The references are too old, only 5 of them are within the most recent three years. To relate your job with the most recent developments can benefit a high impact of your article after publication.
Response: Thank you for recommending us. The new references have been replaced in this submit file.
And the formats are chaotic, upper and lower cases in the references’titles and authors’ names being full or “et al”.
Response: We apologize to the reviewer. In current submit, the reference format has been modified according to the journal's guideline.
Round 3
Reviewer 2 Report
The manuscript can be accepted for publication after correcting one sentence as follows:
The sentence “In addition to the preparation of solid dispersion by evaporation, there are modern techniques that use modern equipment such as electrospun [11] electrospray [12] and fluidized bed technique[13,14], which can control the production factors and/or can be produced in a continuous process.” should be “Both excipients and pharmaceutical techniques are important for effective drug delivery [11] (i.e. Curr Drug Deliv 2021, 18, 2–3). Although the preparation of solid dispersion is reported using modern techniques such as electrospinning [12] electrospraying [14] and fluidized bed technique[14,15], production on a large scale is often difficult for them. (From this standpoint, the merit of your job can be projected.)